



# A high-spatial resolution soil carbon and nitrogen dataset for the
# northern permafrost region, based on circumpolar land cover
# upscaling
Juri Palmtag[1], Jaroslav Obu[2], Peter Kuhry[3, 4], Andreas Richter[5], Matthias B. Siewert[6], Niels Weiss[7];
Sebastian Westermann[2] and Gustaf Hugelius[3,4]
[1]Department of Human Geography, Stockholm University, Stockholm, Sweden; [2]University of Oslo,
Department of Geosciences, Sem Sælands vei 1, 0316 Oslo, Norway; [3]Department of Physical Geography,
Stockholm University, Stockholm, Sweden; [4]Bolin Centre for Climate Research, Stockholm University,
Stockholm, Sweden; [5] Centre for Microbiology and Environmental Systems Science, University of Vienna,
Vienna; [6]Department of Ecology and Environmental Science, Umeå University, Umeå, 901 87, Sweden;
[7]Northwest Territories Geological Survey, Government of the Northwest Territories, Yellowknife NT X1A
1K3, Canada.
Corresponding author: Juri Palmtag (juri.palmtag@humangeo.su.se)





**Abstract**

Soils in the northern high latitudes are a key component in the global carbon cycle; the northern permafrost region covers 22% of the Northern Hemisphere and holds almost twice as much carbon as the atmosphere. Permafrost soil organic matter stocks represent an enormous long-term carbon sink which is in risk of switching to a net source in the future. Detailed knowledge about the quantity and the mechanisms controlling organic carbon storage is of utmost importance for our understanding of potential impacts of and feedbacks on climate change. Here we present a geospatial dataset of physical and chemical soil properties calculated from 651 soil pedons encompassing more than 6500 samples from 16 different study areas across the northern permafrost region. The aim of our dataset is to provide a basis to describe spatial patterns in soil properties, including quantifying carbon and nitrogen stocks, turnover times, and soil texture. There is a particular need for spatially distributed datasets of soil properties, including vertical and horizontal distribution patterns, for modelling at local, regional or global scales. This paper presents this dataset, describes in detail soil sampling, laboratory analysis and derived soil geochemical parameters, calculations and data clustering. Moreover, we use this dataset to estimate soil organic carbon and total nitrogen storage estimates within the soil area of the northern circumpolar permafrost region ($17.9 \times 10^6$ km$^2$) using the ESA's Climate Change Initiative (CCI) Global Land Cover dataset at 300 m pixel resolution. We estimate organic carbon and total nitrogen stocks on a circumpolar scale (excluding Tibet) for the 0-100 cm and 0-300 cm soil depth to 380 Pg and 813 Pg for carbon and 21 Pg and 55 Pg for nitrogen, respectively. Of which 48% of the area is within the land cover class forest with a total SOC and TN storage for 0-300 cm of 35% and 36%, respectively. Our organic carbon estimates agree with previous studies, with most recent estimates of 1000 Pg (– 170 to +186 Pg) to 300 cm depth but show different spatial patterns. Two separate datasets are freely available on the Bolin Centre Database repository (https://doi.org/10.17043/palmtag-2022-pedon-1, Palmtag et al., 2022a and https://doi.org/10.17043/palmtag-2022-spatial-1, Palmtag et al., 2002b).

## 1. Introduction

Permafrost soils represent a large part of the terrestrial carbon reservoir and form a significant and climate-sensitive component of the global carbon cycle (Hugelius et al., 2014). High-latitude ecosystems are experiencing rapid climate change causing warming of soil temperatures, thawing of permafrost, and fluvial and coastal erosion (Biskaborn et al., 2019; Fritz et al., 2017). Warming enhances the decomposition of organic matter by microorganisms, which in turn produces carbon dioxide, methane, and nitrous oxide. The release of these greenhouse gases to the atmosphere is accelerating and could potentially constitute a positive feedback on global warming (Turetsky et al., 2020). To better predict the magnitude and effect of environmental changes in the permafrost region, improved data on the properties and quantities of carbon and nitrogen stored in these climate vulnerable soils are needed.

In many cases, a lack of observational data for parameterization or evaluation can limit model development or accurate model projections (Flato, 2011). Soil properties such as organic matter (OM) content, soil texture and soil moisture or their derivatives are commonly used to parametrize, train or validate models (e.g. Oleson et al., 2010). Yet, the



representation of northern soil profiles in global datasets remains limited (Köchy et al., 2015; Batjes, 2016), the
northern circumpolar permafrost region ($20.6 \times 10^6 \, km^2$) in which permafrost can occur accounts for 22% of the
Northern Hemisphere exposed land area (Obu et al, 2019).
Many previous studies have shown a robust relationship between land cover and soil organic carbon (SOC)
distribution, making land cover datasets useful for upscaling from soil profiles to full landscape coverage (e.g. Kuhry
et al., 2002; Hugelius, 2012; Palmtag et al., 2015; Siewert et al., 2015; Wojcik et al., 2019). Here we describe the
compilation of a harmonized soil dataset for permafrost-affected landscapes derived from 15 different high latitude
sites and one high alpine study site in Canada, Greenland, Svalbard, Sweden, and Russia (Fig. 1; Table 1). In total,
651 soil pedons contain information from up to 6529 samples on carbon and nitrogen content, carbon to nitrogen
(C/N) ratio, isotopic composition, texture (sand, silt+clay) and coarse fraction content, land cover type, wet and dry
bulk density, calculated volumetric contents for ice/water, and volumetric content or organic soil material, mineral
soil material and air. In addition, soil pedon descriptions include metadata on actual sampling site, coordinates and
elevation, slope and aspect, drainage, cover stones and boulders, landform, and maximum sampling depth. Site data
was upscaled to the northern circumpolar permafrost region using the European Space Agency (ESA) Climate Change
Initiative (CCI) Global Land Cover dataset at 300 m pixel resolution, which is the very first long-term global land
cover time series product.
This study has two main objectives. Firstly, the core objective of this dataset is to provide a harmonized, high
resolution, quality controlled, and contextualized soil pedon dataset with a focus on SOC, nitrogen and other
parameters essential to determine the role of northern permafrost region soils in the climate system. Secondly, to use
the dataset and an existing spatial product for upscaling to provide a new and independent estimate of the soil organic
carbon and total nitrogen (TN) storage estimates within the northern circumpolar permafrost region. The data set aims
to provide the scientific community with new and improved geospatial products quantifying carbon and nitrogen pools
within the northern circumpolar permafrost region. Particularly, the extensive metadata on soil properties included for
many samples when available (texture, volumetric densities, active layer depth, ice content, isotopic composition, etc.)
are of great importance and can be used to identify and model the processes responsible for the current and future
carbon balance.

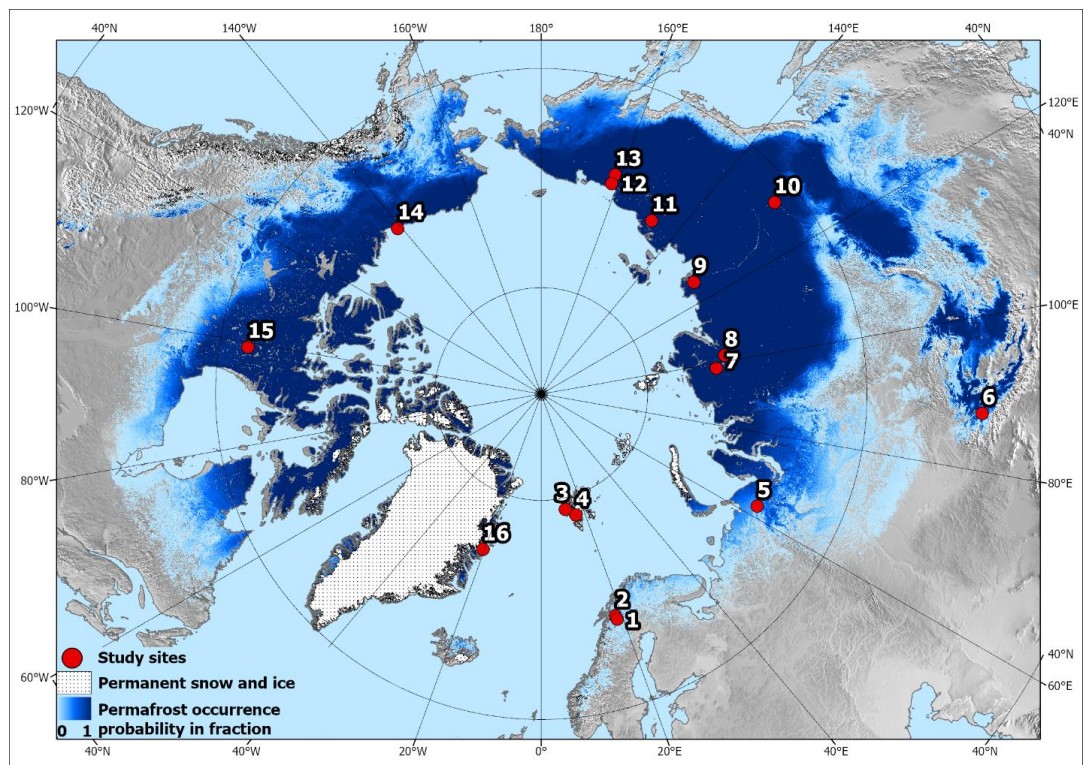

Figure 1: Overview map with location of the 16 sampling sites (see Table 1). Blue shading indicates permafrost probability (dark hues showing higher permafrost occurrence probability), based on an equilibrium state model for the temperature at the top of the permafrost (TTOP) for the 2000–2016 period (Obu et al., 2019). North Pole Lambert azimuthal equal area projection (datum: WGS 84). Base map: Made with Natural Earth.

## 2. Dataset structure

The dataset contains 6529 analyzed samples from 651 soil pedons in 16 different sampling locations across the northern permafrost region (Fig 1; Table 1) (Palmtag et al., 2022a, b). Each sampled pedon was described and classified according to land cover type. Land cover is defined as the biophysical cover of the Earth's terrestrial surface such as different vegetation types, water, and bare ground.



Table 1: Summary of all study sites

| Nr. | Study area | Country | Long | Lat | n=pedons | Reference |
|---|---|---|---|---|---|---|
| 1 | Tarfala | Sweden | 18.63 | 67.91 | 55 | Fuchs et al., 2015 |
| 2 | Abisko | Sweden | 18.05 | 68.33 | 125 | Siewert, 2018 |
| 3 | Ny Ålesund | Norway | 11.83 | 78.93 | 28 | Wojcik et al., 2019 |
| 4 | Adventdalen | Norway | 16.04 | 78.17 | 48 | Weiss et al., 2017 |
| 5 | Seida, Usa River Basin | Russia | 62.55 | 67.35 | 44 | Hugelius et al., 2009; 2011 |
| 6 | Aktru, Altai mountains | Russia | 87.47 | 50.05 | 39 | Pascual et al., 2020 |
| 7 | Logata, Taymyr | Russia | 98.42 | 73.43 | 31 | Palmtag et al., 2016 |
| 8 | Arymas, Taymyr | Russia | 101.90 | 72.47 | 35 | Palmtag et al., 2016 |
| 9 | Lena Delta | Russia | 126.22 | 72.28 | 56 | Siewert et al., 2016 |
| 10 | Spasskaya Pad | Russia | 129.46 | 62.25 | 33 | Siewert et al., 2015 |
| 11 | Tjokurdach | Russia | 147.48 | 70.83 | 27 | Siewert et al., 2015; Weiss et al., 2016 |
| 12 | Shalaurovo | Russia | 161.55 | 69.32 | 22 | Palmtag et al., 2015 |
| 13 | Cherskiy | Russia | 161.30 | 68.45 | 15 | Palmtag et al., 2015 |
| 14 | Herschel Island | Canada | -139.09 | 69.58 | 42 | Siewert et al., 2021 |
| 15 | Tulemalu Lake | Canada | -99.16 | 62.55 | 16 | Hugelius et al., 2010 |
| 16 | Zackenberg | Greenland | -20.50 | 74.45 | 35 | Palmtag et al., 2015; 2018 |



Land cover products are commonly satellite derived and sometimes globally available. We opted for a two-tier
approach, where more classes can be used in products with higher thematic or spatial resolution (Table 2). First, we
differentiated land cover into 7 primary tier classes (Tier I) which represent the major land cover types: Forest, Tundra,
Wetland, Water, Barren, Permanent Snow/Ice and Yedoma. Although Yedoma is a sedimentary deposit and not a
typical land cover class, it was added due to its large areal extent, special soil organic matter (SOM) and ground ice
properties, as well as soil characteristics (Strauss et al., 2017; Weiss et al., 2016). Subsequently, Tier I classes were
subdivided into 14 Tier II subclasses (Table 2). The two-class tier structure provides more detailed information for
each specific land cover class. Depending on the accuracy of the land cover data available for specific sampling sites,
the best fitting Tier level can be used.




Table 2: Hierarchical structure of the two-tier land cover class system applied to the pedons based on field
observations.

| TIER I | | TIER II | |
|---|---|---|---|
| 1 | Forest | 1.1 | Deciduous broadleaf forest |
| | | 1.2 | Evergreen needleleaf forest |
| | | 1.3 | Deciduous needleleaf forest |
| 2 | Tundra | 2.1 | Shrub tundra |
| | | 2.2 | Graminoid / forb tundra |
| 3 | Wetland | 3.1 | Permafrost wetlands |
| | | 3.2 | Non-permafrost wetlands |
| 4 | Water bodies | 4.1 | Lakes |
| | | 4.2 | Streams |
| 5 | Barren | 5.1 | Barren |
| 6 | Snow / Ice | 6.1 | Snow / Ice |
| 7 | Yedoma | 7.1 | Yedoma tundra |
| | | 7.2 | Yedoma forest |

**2.1 Class definitions of soil pedons to land cover types**
All pedons were assigned to land cover classes based on field observations and photographs. The forest class was used
for sparse to dense forests, further separated into three different Tier II classes: deciduous broadleaf, evergreen
needleleaf and deciduous needleleaf forest. Tundra is separated in Tier II to shrub tundra (dominated by erect shrubs
>50cm height) and graminoid / forb tundra (with low growth heath vegetation or graminoid dominated). Wetland
includes terrain that is saturated with water for sufficient time of the year to promote aquatic soil processes with low
oxygen conditions and occurrence of vegetation fully adapted to these conditions, as well as all types of peatlands.
The following types of wetlands described in the field were included to the wetland class: organic wetland, mineral,
seasonal, permanent, ombrogenous and minerogenous mires. Tier II wetland classes are wetlands with permafrost
within the upper 2 m from the soil surface and wetlands without permafrost within the upper 2 m from the soil surface.
Although a substantial part of the northern circumpolar permafrost region is classified as water ($0.98 \times 10^6$ km$^2$) or
permanent snow/ice ($0.06 \times 10^6$ km$^2$), no soil sample or pedon data from these classes are included in the database.
For the same reason the Tibetan permafrost region was not included in our estimates. The class barren includes land
cover types such as exposed bedrock, boulder fields, talus slopes, debris cones, rock glaciers, where soil is either



completely absent, or occurs only in minor patches (<10% area) or in between boulders. The land cover class Yedoma
is defined as areas underlain by late Pleistocene ice-rich syngenetic permafrost deposits, which occupy an area of
about 1,000,000 km$^2$ in Siberia, Alaska, and Yukon (Strauss et al., 2017). Tier II divides the Yedoma domain into
Yedoma tundra and Yedoma forest.

**2.2 Soil sampling and soil analyses**

The main aim of the field studies compiled in the current dataset was to perform SOC/TN pool inventories of each
study area considering different land cover types, geomorphological landforms and soil properties. Field soil sampling
took place in summer months (late June to early September) between 2006 and 2019, most frequently in August or
September in order to capture the maximum seasonal thaw (active layer) depth at each site. At most sites, a stratified
sampling scheme consisting of linear transects with predefined equidistant intervals of typically 100 to 200 m across
all major landscape elements was used. To ensure that this sampling scheme covered all representative landscape units
and types, maps (including vegetation, surficial geology) and remote sensing products (including air photos, satellite
imagery, and elevation models) were assessed prior to fieldwork. Detailed field reconnaissance involving visual
observation of the manageable study area were conducted before establishing transects. Sampling sites were located
and marked at the exact position based on distance to the first sampling point and compass bearing using a hand-held
GPS device. This ensured an unbiased location of individual sampling sites. For some locations, when sufficient time
was available in the field, a random or stratified random distribution of sampling points was used. Following the field
sampling protocol (Figure S1), a site description, soil and in several cases phytomass sampling were conducted at each
sampling point.
For each pedon, the top organic layer was sampled in three replicates, the active layer was sampled from an open soil
pit and deeper sections normally using a steel pipe for soil coring in permafrost. When possible, samples were also
collected from exposures along lake shores or river valleys (Fig. 2). Accurate determination of soil bulk density (BD)
is crucial when converting sample weight to volume or area and is essential to calculate SOC stocks. Therefore, special
attention was paid to accurate soil volume estimation during field sampling. The target depth for soil cores was 100
cm, or until bedrock or massive ground ice (e.g. ice-wedges) was reached. Pedons were occasionally extended beyond
100 cm depth, in particular to assess full peat depth and organic/mineral transition in organic soils.
The top organic layer samples were cut out as a block using a pair of scissors or a knife (removing living vegetation),
measuring the block volume in the field (Fig. 2). Variation in the top organic layer thickness can be substantial and
for this reason from most pedons two randomly selected replicates (OL2 & OL3) in addition to the main soil pit were
collected (not in peatlands). Active layer samples were collected from a soil pit excavated to the bottom of the active
layer, to the bedrock or to reach a depth of ±50 cm, or in a few cases from natural exposures using 100 cm$^3$ soil
sampling rings inserted horizontally into the soil profile. Sampling of the active layer was performed in fixed depth
intervals or along soil horizon boundaries. During some field campaigns, emphasis was also given to the spatial
distribution of soil horizons in the soil pit using perspective corrected photographs to calculate the respective area



covered by each horizon, which was then translated into depth increments (Siewert et al, 2016; 2021). For permafrost-
free wetland sites a Russian peat corer with a 50 cm long chamber was used. After extraction, the core was described
and subdivided into smaller increments (generally 5 cm). This resulted typically in about 5–15 samples per sampling
site depending on the reached depth.
The permafrost section of the soil profile and very deep unfrozen soil layers were sampled using a steel pipe that was
hammered into the frozen ground in short (5 to 10 cm) depth increments. The pipe was pulled out after each sampled
increment using large pipe wrenches, and the sample was pushed out of the pipe using a steel rod. These steel pipes
are industry standardized with an outer diameter of 42.2 mm (1.25 inches), affordable and widely available even in
remote locations. A custom-made protective hard-steel cap placed over the steel pipe greatly extents the usability of
the pipe and this method in general. Over time, pipe-ends deform from hammer impacts and hard objects such as
rocks, and damaged ends can be cut off in the field using a hacksaw. An experienced team of 2-3 persons can sample
4-6 soil pedons in one day using this method. At several locations, soil cores were collected using a handheld
motorized rotational Earth Auger (Stihl BT 121) with a 50 cm core barrel and 52 mm outside diameter. Following
recovery, samples were split lengthwise in preparation for analysis. Usually, half of each core was kept as a frozen
archive to be used in the event of laboratory error. The remaining half-core was analyzed to determine sediment
characteristics, volumetric ice content, and gravimetric water content. Disturbance material was removed from the
core surface by repeated scraping with a razor blade. All half-cores were then photographed and described in detail.
All samples were described in the field and packed into sampling bags. Wet or frozen samples were placed in double
bags to assure no soil water was lost in transport. For each sampled soil profile, pictures and notes were taken to
describe land cover type, landform, elevation, slope and aspect, surface moisture, and surface features. Specific
observations regarding the collected sample depths, such as excess ground ice (visual estimate, %), occurrence of
large stones (visual estimate, %), colour (general description or using a Munsell scale), soil structure, including signs
of cryoturbation, roots and rooting depth were noted. Samples with cryoturbated soil material were marked or rated
on a scale from 1 to 3 according to the relative amount of cryoturbated soil material. Soil texture, which refers to
particle size and relative content of mineral components (sand, silt+clay) is of importance as it affects the physical
and chemical properties of a soil, including cryoturbation (Palmtag and Kuhry, 2018). Soil texture was estimated for
most samples using manipulation tests and assessment by hand in the field under varying weather conditions.
Therefore, we decided to combine silt and clay to avoid misinterpretation and refer to them as one fine-grained soil
texture class. For a subset of samples, particle size analysis was performed using a Malvern Mastersizer 3000 laser
particle size analyzer (Malvern Instruments Ltd, Malvern, UK), which can analyze particles in the range of 0.01–3500
µm in diameter. It measures the intensity of light scattered as a laser beam passes through a dispersed particulate
sample. A detailed description of these samples is given in Palmtag et al. (2018). For some studies soils were classified
following the US Soil Taxonomy system (Soil Survey Staff, 2014). The availability of these parameters is consistent
for most pedons, but the degree of metadata completeness depends on the scope of the original study. The land cover
and vegetation community was described at all sites. For many sites, vegetation cover was described in terms of
relative plant functional type coverage per square meter. Beyond assigning the profiles to land cover, vegetation data
is not included in this database and not further discussed.

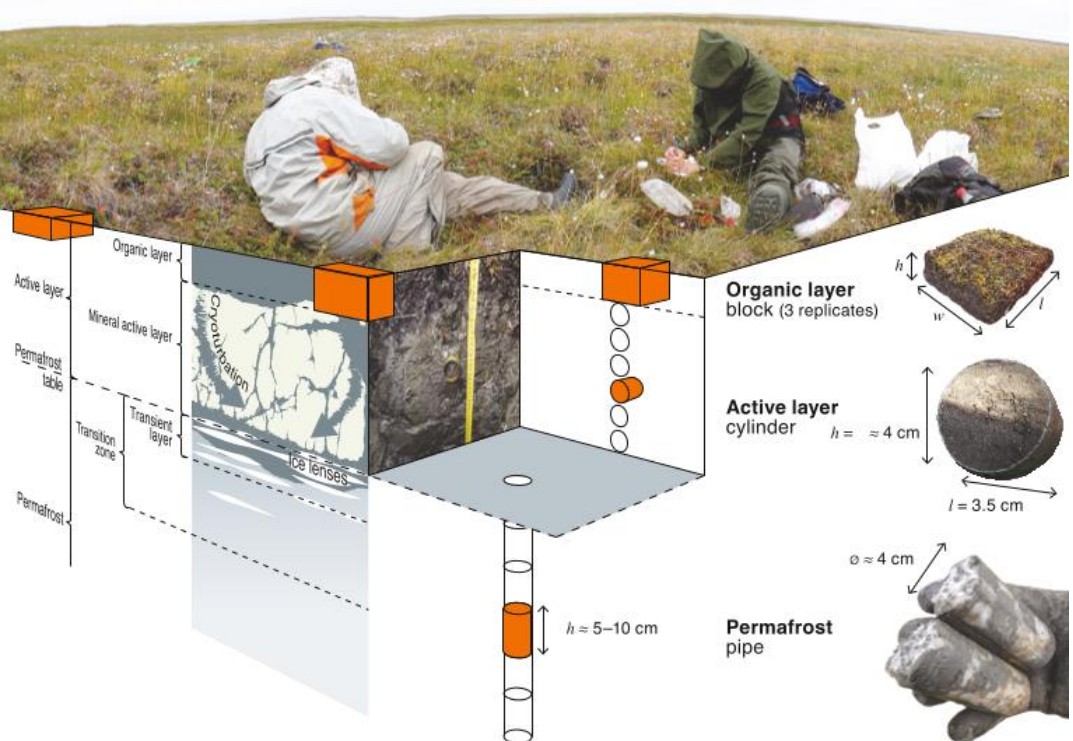


Figure 2: A three-dimensional field sampling protocol with typical soil layers in permafrost ground (reprinted from
Weiss 2017, p.12). The orange shapes represent the different sampling techniques for organic surface layer (block),
active layer sample from an excavated pit (fixed volume cylinder) and permafrost sampling (steel pipe).
**2.3 Laboratory analysis**
In the laboratory, soil samples (n=6529) were weighed before and after oven-drying at 60-70°C for at least 24h (or
until no further weight change was observed) to determine field-moist mass ($m_{ws}$), oven-dried mass ($m_d$), wet bulk
density ($BD_w$) and dry bulk density (BD, g cm$^{-3}$). In most cases, subsamples of around 10g were dried again at 105°C
to verify dry weight and correct in case not all water was lost at the lower temperature. The reason for the main sample
being dried at lower temperature is to ensure that samples can be dried in the original plastic sample bags (without
loss of sampled materials) and subsequently used for additional analyses that may be sensitive to the higher drying
temperature (results from such additional analyses are not included here). After drying, samples were homogenized
and sieved to determine the concentration of coarse mineral fragments (CF,>2 mm, %). Subsamples (n=4488) were
burned for 5h at 550°C to obtain organic matter content through loss on ignition (LOI; Heiri et al., 2001), and about





every second sample (n=2976), was burned at 950°C for 2h to determine carbonate content (for details, see Palmtag
et al., 2015; 2016). To determine the elemental content of carbon and nitrogen (TOC and TN) and their isotopic
composition, 2496 samples were analysed using an Elemental Analyser (EA). If LOI indicated presence of inorganic
carbon, samples were acid treated with hydrochloric acid prior to determination of TOC. To estimate the organic
carbon % for samples where only LOI was available, a regression was performed between LOI550 and %C from EA
on samples for which both analyses were available. In most cases a third or fourth order polynomial regression model
was used and applied at study area level.
Carbon to nitrogen (weight) ratios are often used as an indicator for SOM decomposition. As during the metabolic
activity by microorganisms more carbon than nitrogen is released, the C/N ratio decreases with a higher degree of
humification. This is why C/N ratios usually decrease with depth, as deeper layers are typically older and underwent
decomposition over longer periods of time (Kuhry and Vitt, 1996). Together with stable carbon isotopes (δ13C) this
can be used to gain insight into the biochemical processes of SOM, botanical origin with depth and the degradation
state (Kracht and Gleixner, 2000).
**2.4 Calculations and soil profile extrapolations**
Dry and wet bulk density (g cm$^{-3}$), sample volume (cm$^3$) and % carbon was used to calculate the volumetric contents
of water, organic soil material, mineral soil material and air for each sample. The soil organic carbon content (kgC
m$^{-2}$) was calculated for each sample separately based on dry bulk density (BD, g cm$^{-3}$), percentage organic C in the
sample (%C), sample thickness T (cm), and coarse fraction correction (CF) (Equation 1). Equation 1 was also used to
calculate the TN content, in which %C was replaced with %N.
$$SOC(gCcm^{-2}) = BD * \%C * (1 - CF) * T \qquad\qquad 1$$
SOC content for each pedon was calculated by summing up individual samples to a specific depth (30 cm, 50 cm, 100
cm, 100-200 cm, 200-300 cm, and 0-300 cm) until the maximum sampling depth was reached. In areas with large
stones in the soil column (e.g. alpine areas) or areas with massive ice bodies (e.g. Yedoma deposits), it is also important
to deduct the volume of stones or massive ice from the calculations. These additional variables are not included in
equation 1, but were accounted for in the SOC calculations at the pedon level. If bedrock was encountered at any
point, a SOC content of 0 kg C m$^{-2}$ was assigned for the remaining part down to 300 cm depth at that specific sampling
site. For calculations, the top organic layer calculation is based on the first OL1 sample only. In pedons where some
increments were missing or the full sampling depth was not reached, the nearest samples from the same pedon for BD
and %C were interpolated or extrapolated. To avoid overestimation of the SOC storage, such extrapolations were only
used where field notes showed that the deposits were homogeneous and bedrock was not reached.
Masses of soil components (water ($m_w$, g), organic matter ($m_{OM}$, g) and mineral component ($m_{min}$, g)) were calculated
from laboratory results. The mass of water was calculated as a difference between field-moist mass and oven-dried
mass. Organic matter mass was calculated from the %C and dry sample weight and multiplied by 2, which is a standard



conversion factor between SOC and SOM (Pribyl, 2010). The mass of the mineral fraction was calculated as a
difference between dry sample mass and organic matter mass.
Volumetric fractions of soil components were calculated by dividing the volume of the component with the total
sample volume (V). We calculated component volumes from mass by assuming the following densities: 1 g/cm3 for
water, 0.91 g/cm3 for ice, 1.3 g/cm3 for organic matter (Farouki, 1981) and 2.65 g/cm3 mineral component. The
volumetric fraction of air was calculated as one minus the sum of the other fractions.

### 250 2.5 Pedon grouping and SOC/TN upscaling

All profiles were assigned to land cover class based on field descriptions. Dry bulk density, SOC density, TN density
and the volumetric contents of mineral and organic matter and water and air were averaged according to land cover
classes for depths until 3 m using Python scripting language and pandas library (McKinney, 2011). Soil parameters
were assigned to pedon sample depth ranges and these were grouped according to land cover classes yielding means
and standard deviations for each centimetre of depth. Fractions of soil texture classes (sand and silt+clay) were created
using the same procedure by counting occurrences of texture classes within pedons. The values were averaged with 1
cm resolution for the top 10 cm, to 5 cm between 10 and 30 cm and to 10 cm averaged values for below 30 cm of soil
depth. Typical soil stratigraphies were generated for each class which can be used as input for permafrost modelling
and mapping (e.g. Westermann et al., 2013; 2017; Czekirda et al., 2019).
For the upscaling, we used the land cover map from the Global ESA Land cover Climate Change Initiative (CCI)
project at 300 m spatial resolution (http://maps.elie.ucl.ac.be/CCI/viewer/index.php). The overall classification
accuracy, based on 3167 random sampling cases, is stated as 73% (Defourny et al., 2008). The land cover class dataset
for upscaling was generated from ESA CCI land cover yearly products from period 2006 to 2015 (corresponding to
the sampling period) using majority statistics to define prevailing land cover classes within this period. The extent of
the Yedoma land cover classes was defined from shapefiles of the Yedoma database by Strauss et al., (2016), where
all the layers were used except for QG2500k, which is showing the lowest probability of Yedoma occurrence. While
the extent of Yedoma is based on maps of Quaternary deposits, no such maps were used to support upscaling of other
soil properties.
The spatial land cover extent was constrained to the Northern Hemisphere permafrost region indicating probability of
permafrost occurrence but not the actual area underlain by permafrost (indicated by permafrost area) (Obu, 2021). The
used permafrost region dataset stretches over $17.9 \times 10^6$ km$^2$ of the Northern Hemisphere, and is based on equilibrium
state model for the temperature at the top of the permafrost (TTOP) for the 2000–2016 period (Obu et al., 2019).
Since the ESA land cover product uses a different nomenclature for land cover types with different sub-categories,
similar classes were amalgamated to fit our tiered land cover system (Table 2). Several minor classes consisting of
single pixels spread over the map were generalized and merged with the class surrounding the pixel. The Tier classes
"water bodies" and "Snow / Ice" occupy substantial areas but were excluded from the SOC storage estimates; this



study focuses on terrestrial SOC and TN storage. We defined tier II Yedoma classes (Yedoma tundra and Yedoma
forest) according to the ESA CCI Land cover classes coinciding with Yedoma deposits (Table 3).

Table 3: Amalgamation of ESA's CCI land cover classes with the Tier class system above the Yedoma deposits.

| CCI class | ESA CCI landcover | TIER I class | TIER II class |
|---|---|---|---|
| 40 | Mosaic natural vegetation (tree, shrub, herbaceous cover) (>50%) | 1 | 1.1 & 7.2 |
| 50 | Tree cover, broadleaved, evergreen, closed to open (>15%) | 1 | 1.1 & 7.2 |
| 60 | Tree cover, broadleaved, deciduous, closed to open (>15%) | 1 | 1.1 & 7.2 |
| 61 | Tree cover, broadleaved, deciduous, closed (>40%) | 1 | 1.1 & 7.2 |
| 70 | Tree cover, needleleaved, evergreen, closed to open (>15%) | 1 | 1.2 & 7.2 |
| 71 | Tree cover, needleleaved, evergreen, closed (>40%) | 1 | 1.2 & 7.2 |
| 72 | Tree cover, needleleaved, evergreen, open (15-40%) | 1 | 1.2 & 7.2 |
| 80 | Tree cover, needleleaved, deciduous, closed to open (>15%) | 1 | 1.3 & 7.2 |
| 90 | Tree cover, mixed leaf type (broadleaved and needleleaved) | 1 | 1.1 & 7.2 |
| 100 | Mosaic tree and shrub (>50%) / herbaceous cover (<50%) | 1 | 1.1 & 7.2 |
| 110 | Mosaic herbaceous cover (>50%) / tree and shrub (<50%) | 1 | 1.3 & 7.2 |
| 120 | Shrubland | 2 | 2.1 & 7.1 |
| 121 | Evergreen shrubland | 2 | 2.1 & 7.1 |
| 122 | Deciduous shrubland | 2 | 2.1 & 7.1 |
| 130 | Grassland | 2 | 2.2 & 7.1 |
| 140 | Lichens and mosses | 2 | 2.2 & 7.1 |
| 150 | Sparse vegetation (tree, shrub, herbaceous cover) (<15%) | 2 | 2.1 & 7.1 |
| 152 | Sparse shrub (<15%) | 2 | 2.1 & 7.1 |
| 160 | Tree cover, flooded, fresh or brackish water | 3 | 3.1 |
| 180 | Shrub or herbaceous cover, flooded, fresh/saline/brackish water | 3 | 3.1 |
| 200 | Bare areas | 5 | 5.1 |
| 201 | Consolidated bare areas | 5 | 5.1 |
| 202 | Unconsolidated bare areas | 5 | 5.1 |
| 210 | Water bodies | 4 | 4.1 |
| 220 | Permanent snow and ice | 6 | 6.1 |


The upscaling to estimate the total carbon storage in the northern circumpolar permafrost region was performed in
ArcGIS Pro (ESRI, Redlands, CA, USA) by multiplying the mean SOC storage for each tier and tier 2 class with the
spatial extent of the corresponding CCI land cover class. To determine reasonable error estimates for carbon stocks
within the permafrost region, we used a spatially weighed 95% confidence interval (CI) as described by Thompson
(1992) assuming that our residuals are normally distributed (Hugelius, 2012). The CI accounts for the relative spatial
extent, carbon stock variations in pedons and number of replicates in each upscaling class. Replicates were only
considered for pedons reaching the full depth, resulting in fewer replicates with increasing sampling depth. In equation





2: *t* is the upper α/2 of a normal distribution (t≈1.96), *a* the % of the area; *SD* is the standard deviation, *n* is to the
number of replicates and *i* refers the specific Tier class.
$$CI = t * \sqrt{\Sigma((a_i{}^2 * SD_i{}^2)/n_i)}$$    2
**3. Results**
**3.1 SOC estimates**
Using our pedon based dataset, we obtain SOC stock estimates within the northern circumpolar permafrost region of
379.7 and 812.6 Pg for 0-100 cm and 0-300 cm depth, respectively. Table 4 shows mean SOC storage (kg C/m2) and
total SOC stock for all depth increments, including 95% confidence intervals. The upscaling using this new pedon
data shows that almost half of SOC in the northern circumpolar permafrost region is stored in the top meter. The three
most abundant classes together (deciduous needleleaf forest, shrub tundra and graminoid / forb tundra) occupy 67%
of the permafrost region (Table 5) and store the bulk of terrestrial SOC in the northern circumpolar region (74%). The
permafrost wetland class has the largest SOC content to 300 cm with 112.2 kg C /m$^{-2}$, but has only a small areal
coverage in the ESA LCC product (1.4%) which results in a total SOC storage contribution of 3.5% within the
permafrost region. Figure 3 illustrates the spatial distribution of total SOC storage (kg C m$^{-2}$) to a depth of 300 cm for
the circumpolar permafrost region.
Table 4: Landscape mean and total SOC storage with 95% CI for the different depth increments for the northern
circumpolar permafrost region, excluding water bodies and permanent snow and ice.

| Depth increment | n: | Landscape mean SOC storage (kg C/m²) | 95% CI [a] | | Total SOC in Pg | 95% CI [a] | |
|---|---|---|---|---|---|---|---|
| 0-30 | 452 | 9.0 | ± | 1.4 | 160.0 | ± | 25 |
| 0-50 | 402 | 12.8 | ± | 1.8 | 229.3 | ± | 32 |
| 0-100 | 328 | 21.3 | ± | 3.2 | 379.7 | ± | 58 |
| 100-200 | 257 | 12.4 | ± | 1.9 | 222.0 | ± | 35 |
| 200-300 | 253 | 11.8 | ± | 1.7 | 211.0 | ± | 31 |
| 0-300 | 253 | 45.5 | ± | 7.6 | 812.6 | ± | 136 |


[a] The 95% confidence interval refers to landscape mean SOC storage and total SOC storage



Table 5: Mean and total SOC storage for (A) 0-100 cm and (B) 0-300 cm soil depth separated for the different Tier
classes in the northern circumpolar permafrost region, excluding water bodies and permanent snow and ice.

| A | Tier class | LCC class | n [a]: | Area (million km²) | Area % | Mean SOC storage (kg C/m²) [b] | SD [b] | Total SOC in Pg | Total SOC storage % |
|---|---|---|---|---|---|---|---|---|---|
| | 1.1 | Deciduous broadleaf forest | 5 | 0.85 | 4.8% | 16.5 | 9.3 | 14.1 | 3.7 |
| | 1.2 | Evergreen needleleaf forest | 4 | 2.54 | 14.3% | 14.6 | 12.8 | 37.1 | 9.8 |
| | 1.3 | Deciduous needleleaf forest | 28 | 5.20 | 29.1% | 20.5 | 20.3 | 106.5 | 28.1 |
| | 2.1 | Shrub tundra | 54 | 3.97 | 22.3% | 22.3 | 21.7 | 88.5 | 23.3 |
| | 2.2 | Graminoid / forb tundra | 118 | 2.85 | 15.9% | 31.6 | 23.0 | 90.0 | 23.7 |
| | 3.1 | Permafrost wetlands | 61 | 0.25 | 1.4% | 37.8 | 37.8 | 9.6 | 2.5 |
| | 3.2 | Non-permafrost wetlands | 10 | 0.76 | 4.3% | 17.8 | 14.7 | 13.5 | 3.6 |
| | 5.1 | Barren | 39 | 0.85 | 4.8% | 9.4 | 12.0 | 8.0 | 2.1 |
| | 7.1 | Yedoma tundra | 8 | 0.27 | 1.5% | 28.1 | 17.0 | 7.7 | 2.0 |
| | 7.2 | Yedoma forest | 1 | 0.30 | 1.7% | 16.1 | 0.0 | 4.8 | 1.3 |

| B | Tier class | LCC class | n [a]: | Area (million km²) | Area % | Mean SOC storage (kg C/m²) [b] | SD [b] | Total SOC in Pg | Total SOC storage % |
|---|---|---|---|---|---|---|---|---|---|
| | 1.1 | Deciduous broadleaf forest | 2 | 0.85 | 4.8% | 33.2 | 22.8 | 28.3 | 3.5 |
| | 1.2 | Evergreen needleleaf forest | 2 | 2.54 | 14.3% | 23.0 | 16.3 | 58.7 | 7.2 |
| | 1.3 | Deciduous needleleaf forest | 14 | 5.20 | 29.1% | 38.3 | 33.3 | 199.2 | 24.5 |
| | 2.1 | Shrub tundra | 50 | 3.97 | 22.3% | 49.2 | 50.8 | 195.6 | 24.1 |
| | 2.2 | Graminoid / forb tundra | 114 | 2.85 | 15.9% | 72.2 | 67.5 | 205.4 | 25.3 |
| | 3.1 | Permafrost wetlands | 49 | 0.25 | 1.4% | 112.2 | 121.5 | 28.4 | 3.5 |
| | 3.2 | Non-permafrost wetlands | 7 | 0.76 | 4.3% | 74.5 | 70.5 | 56.6 | 7.0 |
| | 5.1 | Barren | 9 | 0.85 | 4.8% | 11.7 | 14.9 | 10.0 | 1.2 |
| | 7.1 | Yedoma tundra | 5 | 0.27 | 1.5% | 64.1 | 37.7 | 17.5 | 2.2 |
| | 7.2 | Yedoma forest | 1 | 0.30 | 1.7% | 43.0 | 0.0 | 13.0 | 1.6 |


[a] The number of sampled pedons reaching a full depth of 100 cm or 300 cm, respectively.
[b] Mean SOC storage and SD calculations includes pedons which are not reaching the full section depth.







Figure 3. Estimated total SOC storage (kg C m$^{-2}$) to a depth of 0-100 cm and 0-300 cm in northern circumpolar permafrost region. North Pole Lambert azimuthal equal area projection (datum: WGS 84). Base map: Made with Natural Earth.



### 3.2 TN estimates

Our estimates show that the TN stocks down to 100 cm and 300 cm depth in the northern circumpolar permafrost
region are 21.1 Pg and 55.0 Pg, respectively. Table 6 presents the mean and total TN storage for different depth
increments with their 95% confidence interval. The TN distribution throughout the full depth is more evenly
distributed compared to SOC. As with SOC storage, the most abundant land cover classes (deciduous needleleaf forest,
shrub tundra and graminoid / forb tundra) store the bulk (68%) of the total TN in the permafrost region. The land
cover classes permafrost and non-permafrost wetlands have the largest TN storage with a mean of up to 7 kg /N m$^2$
for the 0-300 cm soil depth (Table 7). Figure 4 illustrates the spatial distribution of total TN storage (kg N m$^{-2}$) for the
circumpolar permafrost region for two depth intervals, 0-100 cm and 0-300 cm.
Table 6: Mean and total TN storage with 95% CI for the different depth increments for the northern circumpolar
permafrost region, excluding water bodies and permanent snow and ice.

| Depth increment | | Landscape mean TN storage (kg N/m²) | 95% CI [a] | | Total TN in Pg | 95% CI [a] | |
|---|---|---|---|---|---|---|---|
| 0-30 | 271 | 0.5 | ± | 0.1 | 8.1 | ± | 1.5 |
| 0-50 | 250 | 0.7 | ± | 0.1 | 12.3 | ± | 2.5 |
| 0-100 | 208 | 1.2 | ± | 0.3 | 21.1 | ± | 4.7 |
| 100-200 | 175 | 1.0 | ± | 0.2 | 17.1 | ± | 2.8 |
| 200-300 | 169 | 0.9 | ± | 0.2 | 16.8 | ± | 3.7 |
| 0-300 | 169 | 3.1 | ± | 0.8 | 55.0 | ± | 15.1 |


[a] The 95% confidence interval refers to landscape mean TN storage and total TN storage

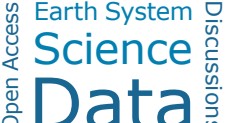

Table 7: Mean and total TN storage for (A) 0-100cm and (B) 0-300 soil depth separated for the different Tier classes
within the northern circumpolar permafrost region, excluding water bodies and permanent snow and ice.

| A | Tier class | LCC class | n [a]: | Area (million km²) | Area % | Mean TN storage (kg N/m²) [b] | SD [b] | Total TN in Pg | Total TN storage % |
|---|---|---|---|---|---|---|---|---|---|
| | 1.1 | Deciduous broadleaf forest | 2 | 0.85 | 4.8% | 1.0 | 0.6 | 0.9 | 4.1 |
| | 1.2 | Evergreen needleleaf forest | 1 | 2.54 | 14.3% | 0.8 | 0.8 | 1.9 | 9.2 |
| | 1.3 | Deciduous needleleaf forest | 19 | 5.20 | 29.1% | 1.0 | 0.6 | 5.1 | 24.3 |
| | 2.1 | Shrub tundra | 32 | 3.97 | 22.3% | 1.6 | 1.5 | 6.4 | 30.3 |
| | 2.2 | Graminoid / forbtundra | 72 | 2.85 | 15.9% | 1.5 | 0.9 | 4.3 | 20.3 |
| | 3.1 | Permafrost wetlands | 46 | 0.25 | 1.4% | 2.4 | 2.5 | 0.6 | 2.8 |
| | 3.2 | Non-permafrost wetlands | 4 | 0.76 | 4.3% | 0.7 | 0.6 | 0.5 | 2.4 |
| | 5.1 | Barren | 26 | 0.85 | 4.8% | 0.7 | 0.9 | 0.6 | 2.6 |
| | 7.1 | Yedoma tundra | 5 | 0.27 | 1.5% | 1.6 | 0.6 | 0.4 | 2.0 |
| | 7.2 | Yedoma forest | 1 | 0.30 | 1.7% | 1.4 | 0.0 | 0.4 | 2.0 |

| B | Tier class | LCC class | n [a]: | Area (million km²) | Area % | Mean TN storage (kg N/m²) [b] | SD [b] | Total TN in Pg | Total TN storage % |
|---|---|---|---|---|---|---|---|---|---|
| | 1.1 | Deciduous broadleaf forest | 2 | 0.85 | 4.8% | 2.8 | 1.7 | 2.4 | 4.3 |
| | 1.2 | Evergreen needleleaf forest | 1 | 2.54 | 14.3% | 1.9 | 2.3 | 4.8 | 8.8 |
| | 1.3 | Deciduous needleleaf forest | 12 | 5.20 | 29.1% | 2.4 | 1.3 | 12.6 | 23.0 |
| | 2.1 | Shrub tundra | 30 | 3.97 | 22.3% | 3.9 | 3.4 | 15.5 | 28.2 |
| | 2.2 | Graminoid / forbtundra | 69 | 2.85 | 15.9% | 3.4 | 2.2 | 9.6 | 17.5 |
| | 3.1 | Permafrost wetlands | 40 | 0.25 | 1.4% | 7.0 | 7.8 | 1.8 | 3.2 |
| | 3.2 | Non-permafrost wetlands | 2 | 0.76 | 4.3% | 6.4 | 6.6 | 4.9 | 8.9 |
| | 5.1 | Barren | 9 | 0.85 | 4.8% | 0.8 | 1.1 | 0.7 | 1.2 |
| | 7.1 | Yedoma tundra | 3 | 0.27 | 1.5% | 5.6 | 2.2 | 1.5 | 2.8 |
| | 7.2 | Yedoma forest | 1 | 0.30 | 1.7% | 4.1 | 0.0 | 1.2 | 2.2 |


[a] The number of sampled pedons reaching a full depth of 100 cm or 300 cm, respectively.
[b] Mean TN storage and SD calculations includes pedons which are not reaching the full section depth.



Figure 4. Estimated total Nitrogen storage (kg C m⁻²) to a depth of 0-100 cm and 0-300 cm in northern circumpolar

permafrost region. North Pole Lambert azimuthal equal area projection (datum: WGS 84). Base map: Made with

Natural Earth.



### 3.3 Typical vertical soil stratigraphies to 300 cm depth

Figure 5 illustrates averaged vertical soil stratigraphies for SOC and TN density, C/N ratio with d13C, dry bulk density, volumetric fractions for water/ice, organic, mineral, air and texture (sand, silt+clay fraction) separated by land cover class to 300 cm depth. The data shows clear differences occurring in the more variable top meter in comparison to the rather stable second and third meter. These important trends are more evident, e.g. high variability in water fraction between classes or high silt+clay fraction, in Yedoma tundra. We note that mineral soil texture is mainly determined by the parent material origin, which has not been accounted for in the generation of these profiles.

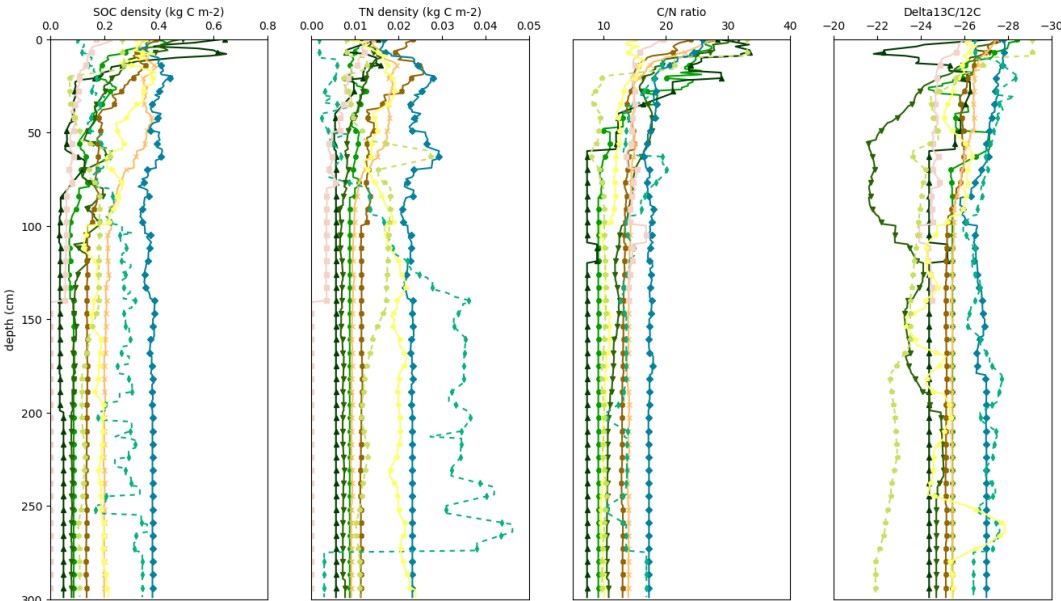

351

352

353    Figure 5. Typical vertical soil stratigraphies for all the land cover classes to 300 cm depth



## 4. Discussion

This new open access database was created to serve different scientific communities with high spatial resolution, harmonized and georeferenced data on soil organic carbon and related key pedological parameters representative for large areas across the northern permafrost region. Georeferenced and quality assessed soil profile data, with extensive metadata will allow users to relate this field-data to various other ecosystem properties or processes. The goal of the field studies to collect this dataset has mainly been to improve the knowledge base for studies of climate feedbacks resulting from permafrost thaw. While there are multiple databases available containing data on soil carbon storage (Hugelius et al., 2013, Michaelson et al., 2013, Mishra et al., 2021), there is still a lack of soil field data covering a wider range of properties within the hard-accessible northern circumpolar permafrost region. This database provides detailed high-resolution soil profile data on different key soil properties for both chemical (organic carbon, total carbon, total nitrogen, d13C) and physical (dry and wet bulk density, soil texture, coarse fragments) parameters.

To test and exemplify usage of the soil profile database, we used our field-based metadata to classify soil profiles according to a coherent land cover scheme and combined it with ESA's land cover product to provide a new estimate of soil organic carbon storage in the northern circumpolar permafrost region. Our estimate for SOC is 380 Pg ± 58 Pg to 100 cm soil depth and 813 Pg ± 136 Pg to 300 cm soil depth for the permafrost region occupying an area of $17.9 \times 10^6$ km$^2$ (excluding area of Tibetan permafrost region, permanent snow and ice and water bodies). In comparison, Hugelius et al., (2014) estimated SOC stocks in the northern circumpolar permafrost region ($17.8 \times 10^6$ km$^2$ excluding exposed bedrock, glaciers and ice-sheets and water bodies) to be 472 ± 27 Pg and 1035 ± 150 Pg to 100 cm and 300 cm for soils, respectively. A recent publication by Mishra et al., (2021) based on >2700 soil profiles with environmental variables in a geostatistical mapping framework, estimated a total SOC stock of 510 Pg (– 78 to +79 Pg) and 1000 (– 170 to +186 Pg) to 100 cm and 300 cm, respectively. Despite different approaches in upscaling, with Hugelius et al., (2014) using regional soil maps and Mishra et al., (2021) digital soil mapping, our landcover based estimate is on the lower edge to previous studies. However, our estimates are still within each other's error estimates. In comparison, this upscaling technique offers further benefits as this database can be easily extended with additional sampling sites, higher-resolution land cover maps that will further increase the resolution on a circumpolar scale. This data can also be used for upscaling in a particular area of interest.

Despite the importance of nitrogen for microbial decomposition and plant productivity processes, few large-scale datasets are available on TN storage. Our TN estimate for the northern circumpolar permafrost region is 21 Pg ± 5 Pg to 100 cm soil depth and 55 Pg ± 15 Pg to 300 cm soil depth. This is in line with the only other circumpolar estimate known to use by Harden et al. (2012) with a best estimate of 66 Pg (± 35 Pg).

In addition, C/N ratio is a useful indicator of the organic matter decomposability which usually decreases with depth with least decomposed material at the surface (organic layer) followed by carbon enriched (cryoturbated) pockets and with smallest values and the most degraded material the mineral subsoil. Therefore, the C/N data together with the d13C data locates the areas which are most likely to be more vulnerable to permafrost degradation which can be used as a vulnerability map in combination with the botanical origin of the plant species using carbon isotopes.



A key element to this upscaling exercise is the accuracy of the land cover dataset. Despite the relatively high spatial
resolution of 300m, many Arctic landscape features cannot be represented at this scale. In addition, ESA's land cover
map has a good overall accuracy of 73%; however, this means that 27% of the land cover is possibly mismatched and
in need of improvement. Moreover, the accuracy for natural and semi-natural aquatic vegetation is unfortunately as
low as 19%. This corresponds to the class (wetland) with the largest SOC content in the permafrost region. According
to Hugelius et al. (2020), the areal extent of peatlands for the northern permafrost region ($3.7 \times 10^6$ km$^2$) is almost four
times the ESA's land cover product estimated areal extent ($1.0 \times 10^6$ km$^2$), used in this study. This would partly explain
our throughout lower estimate for SOC and TN on a circumpolar scale since the wetland classes have the largest SOC
and TN contents, particularly at greater depths (100-300 cm). If we correct the wetland area to $3.7 \times 10^6$ km$^2$
($2.0 \times 10^6$ km$^2$ in permafrost-free peatlands and $1.7 \times 10^6$ km$^2$ permafrost-affected peatlands) and deduct this in
proportion from the other classes, our updated SOC and TN stock to 300 cm soil depth increases to 954 Pg ± 162 Pg
and 66 Pg ± 22 Pg, respectively.
To our knowledge, this is the first product which presents different key soil properties and parameters on a circumpolar
scale even though they are the ones that are commonly used to parameterize earth system models. With this database
we aim to provide georeferenced point data that can easily be implemented and used for geospatial analysis at a
circumpolar scale. This will assist to quantify and model ongoing pedological and ecological processes relevant to
climate change. Furthermore, this may help identifying regions that are more vulnerable to permafrost degradation
and greenhouse gas release due to knowledge on texture, water/ice content or SOC storage.
**5. Conclusion**
This dataset represents a substantial contribution of high-quality soil pedon data across the northern permafrost region.
Despite a different methodology, our estimates of total SOC are similar but on the lower edge to other recent numbers.
The lower estimates from our dataset are probably due to underestimated areal extent of northern peatlands in the ESA
Global Land Cover dataset. In addition to SOC data, we contribute with novel TN estimates for the different land
cover classes and depth increments. Our TN estimate to 300 cm soil depth (55 Pg ± 15 Pg) is in line with the only
other product available on that scale. We provide data for a wide range of environments and geographical regions
across the permafrost region including georeferencing and metadata. This serves as a base that can be easily combined
and extended with data from other sources, as several regions are underrepresented (Alaska, Canada, Tibet). This
dataset offers high scientific value as it also contains data on chemical and physical soil properties across the northern
circumpolar permafrost region. This additional data is of high importance and can be used to develop or parametrize
broad scale models and to help better understand different aspects of the permafrost-carbon climate feedback.
**6. Data access**
Two separated datasets (Detailed pedon data on soil carbon and nitrogen for the northern permafrost region,
https://doi.org/10.17043/palmtag-2022-pedon-1) (Palmtag et al., 2022a) and (A high spatial resolution soil carbon and



nitrogen dataset for the northern permafrost region, https://doi.org/10.17043/palmtag-2022-spatial-1) with GeoTiffs
(Palmtag et al., 2022b) are freely available on the Bolin Centre data set repository (https://bolin.su.se/data/).

**Funding**
This study was funded through the European Space Agency CCI + Permafrost project (4000123681/18/I-NB), the
European Union Horizon 2020 research and innovation project Nunataryuk (773421), the Changing Arctic Ocean
(CAO) program project CACOON (NE/R012806/1) and the Swedish Research Council (2018-04516).

**Author contribution**
GH, PK, SW and JP designed the concept of the study. JO wrote the script in Python. JP wrote the initial draft of the
manuscript. All authors contributed to the writing and editing of the manuscript.
**Competing Interests**
The authors declare that they have no conflict of interest.
**Acknowledgements**
We thank the ESA CCI Land Cover project for providing the data, which was used for upscaling our product to
circumpolar scale.

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
