# Peer review of "A high-spatial resolution soil carbon and nitrogen dataset for the"

_Earth System Science Data, 2022_

## Author Response (AR1)

**Dear Reviewer 1 and Reviewer 2.**

**Thank you for your comments which helped to improve our manuscript. Here we respond to the comments point by point:**

**Response to the anonymous Referee #1:**

**Major comments**

The authors present a very large dataset of soil samples from different land cover classes in the permafrost region. They used the data to calculate carbon and nitrogen storage estimates for the northern permafrost region with an upscaling approach. This is a very valuable study and especially the nitrogen storage estimates as these have not been the focus of many studies in permafrost regions. However, there are some things that could be improved to this study which I listed below, and I recommend major revisions of the manuscript before publication.

I agree that it is important to distinguish between Yedoma and non-Yedoma sediments. However, I don't understand why this distinction goes only so far to say that there is Yedoma tundra or forest. Isn't it also important to distinguish between the broadleaf and needle leaf forests within the Yedoma region, and between the shrub and graminoid / forb tundra? Also, I would leave out water bodies and snow/ice out of this table, as you have no samples from these land cover classes, and only mention them in the text instead, just to make the table clearer. Thus, I would propose to introduce Yedoma as a separate level (or tier) of the land cover class system and distinguish between Yedoma or non-Yedoma sediments, then between the four classes (forest, tundra, wetland and barren) and then the corresponding subclasses.

> First of all, thank you very much for your valuable comments. We fully agree, that it is important to distinguish Yedoma in the way we did with the other classes (forest, tundra, wetland and barren) but unfortunately, we have too few sites. As shown in table 5 and 7, we only have 8 pedons to 100 cm in Yedoma tundra and only 1 pedon in Yedoma forest. However, the idea was to create this subdivision and to point out the importance. We also introduced the tier levels, which can be extended beyond tier 2. As mentioned in several places, there are other areas (Canada, Alaska, Tibet, high alpine areas) which are underrepresented. Since we provide all the parameters and coordinates, this dataset can be easily combined and extended with these important but underrepresented areas.

> As suggested, we removed the water bodies and snow/ice classes to make it clearer for the reader.

I think the methods chapter is very long. You could consider to move part of it to the supplements and have a more concise description in the manuscript itself. I am completely missing the description (results) and interpretation (discussion) of the spatial distribution of C and N storage, as well as from the other soil parameters (C/N ratio, δ13C, BD, volumetric fractions, texture). Please incorporate this!

The methods chapter is now shortened. Additional information and interpretation about the spatial distribution of C and N is added in both sections, in the result and the discussion section. The core objective is to provide and describe the dataset and secondly, to quality test same dataset to quantify the carbon and nitrogen pools within the northern circumpolar permafrost region. As described in the methods section, there are many more soil parameters available but are beyond the focus of this "Earth System Science Data" paper.

**Minor comments**

*Abstract*

L34: please rephrase "within the soil area" (for example: "in soils in the northern…") as you report the C and N storage estimates for a volume, not an area.

Changed as suggested

L38: the sentence should not start with "of which"

I removed this sentence as it is not relevant for the abstract.

L40: "but show different spatial patterns" –> this is the only time in the paper that you say anything about the spatial distribution

Thank you. I removed the "different spatial distribution pattern" part from the abstract as we are not actually comparing the pattern here.

L41-43: this is not the right place to cite these datasets

I removed the references and added following as suggested by ESSD: "Dataset references and DOIs are presented in the "Data access" section in the end."

*Introduction*

L47: temperatures can't warm, please rephrase to "warming of the soil" or "increasing soil temperatures"

Thank you for the comment, changed as suggested.

L50: isn't the accelerating you mention part of the feedback? Please rephrase

I rephrased the sentence to "The release of these greenhouse gases to the atmosphere would in turn generate further climate change, resulting in a positive feedback on global warming (Turetsky et al., 2020)."

L54: introduce the abbreviation OM in line 48 instead

Introduced the abbreviation earlier as suggested

L60: I am missing what you are upscaling, maybe you can add the word data or estimates

Added estimates as suggested.

L69: here you refer to data as singular whereas in L51-52 you are referring to data as plural

Changed to plural here as well, thank you.

L72: I think you mean aims here (what you hope to achieve)

> Changed to aims

L76-78: I would not introduce another aim here, you can leave out this sentence

> I removed this sentence.

*Methods*

The subchapters about the sampling, lab analyses and calculations all fall under the main chapter "Dataset structure" which I think is not so fitting. Consider to rename the chapters such as: 2. Methods, 2.1 Dataset structure, 2.1.1 Class definitions of soil pedons to land cover types, 2.2 Soil sampling, 2.3 Laboratory analysis, etc. Also, you could combine chapter 2.4 and 2.5 or make a clearer distinction between the chapters.

> Changes made as suggested: Methods chapter restructured and section 2.4 and 2.5 combined.

A few times you mention "at most sites", "for some locations", "occasionally", "normally", "when possibly", etc. This is not very helpful if it is unclear why you only carried out certain procedures on a subset of the sites and what happened for the other sites. Please explain

> Agree and thankful for the good comment. Methods section restructured and partly rewritten. All the above issues are now hopefully solved.

L101-102: use small letters for the land cover types

> Changes as suggested.

L131-132: you can remove this here

> Removed as suggested

L134-136: this is a bit vague. How many samples were taken in these 100-200 m intervals?

> Additional information added in the text "582 out of 651"

L150: can you indicate here how many soil pedons exceeded 1 m or reached 3 m?

> The number of pedons which extended below 100 cm, is 313. This number is added. In addition, the n for pedons reaching the depth of 100, 200 and 300 is shown in Table 4.

L153: rephrase "measuring the block volume in the field" to "and the block volume was measured in the field"

> Changed as suggested.

L153-154: in L145 you mentioned you took 3 replicates samples of the organic layer but here you say that you took replicates sometimes. Later (L237) you say you only used the first of the three replicates. Why is that? And even later (L287-288), you mention the replicates were only considered for pedons reaching the full depth. Do you here refer to other replicates than the organic layer replicates?

> Thank you for the comment and pointing out the confusion. As explanation, in some cases, 3 replicates were collected for the organic layer due to substantial variation, but these were not used in the SOC/TN calculations. Sections where OL replicates were

mentioned are now removed as they are not part of the calculations and therefore nor relevant in this manuscript.

Comment to the line 287-288: Yes, replicates here refers to sampled pedons. A pedon was only considered suitable if the full depth was reached. Since the OL replicates are now removed from the text, hope this is now more clear.

L158-160: what is the relevance of this information?

Agree and removed as not relevant here.

L160: permafrost-free should be non-permafrost

Changed to non-permafrost

L168-L171: this can be left out

Removed from the manuscript

L172-172: what do you mean with "following recovery"?

The whole part was rewritten to: "Samples were split lengthwise into two halves: one half was analyzed to determine sediment characteristics, volumetric ice content, and gravimetric water content."

L175-176: I think this sentence should be moved up before the lengthwise splitting

Moved up

L188-191: please move this to the chapter "Laboratory analysis" and rename this subchapter "Soil sampling"

As suggested, moved this part to the Laboratory section and renamed the section.

L191-193: I think these sentences are not necessary. If they are, please put them into context

Agree with the reviewer that this part is not giving any necessary information and is therefore removed from the manuscript.

L202-208: this is unclear to me. How did you determine the bulk density with only the weight before and after? Why did you not dry all samples at 60-70 °C and 105 °C so that you can use the samples dried at a low temperature for further analyses and the weight difference from samples dried at a high temperature for the calculations? Was the correction really necessary or in other words, was the weight difference very different for the samples dried at low and high temperatures? If yes, how can you assume that the weight difference is correct for those samples where you did not dry the subsamples at the higher temperature?

Added additional comment that the bulk density was obviously determined from the weight and the volume. Also, comment to the drying procedure. All samples, now with a n of 3684 out of 5230, which were organic rich or fine grained were dried again at 105°C to exclude the possibility of remaining water. Remaining samples which were not dried again, were sand or course grain samples and showed in tests no remaining water.

L211: rephrase "every second sample" to "half of the samples"

Changed

L215: introduce the abbreviation organic C % here

Done

L216-217: why in most cases? What was the alternative?

Changed to: "A third or fourth order polynomial regression model was used…"

L221: write 13 in δ13C in superscript

Changed to superscript throughout the manuscript

L242: explain "from laboratory results" better

Changed to: "were calculated based on the laboratory analysis for all the individual samples." How it was calculated is described below.

L256-258: in line 231 you mention different intervals. Why did you average the values with a 1 cm resolution if you use 1 value per depth interval for the actual calculations?

Thank you for pointing out the confusion. Our data is on 1 cm resolution. But the depth increments are used for comparison with other publications. Text now changed in the manuscript to: "*SOC content for each pedon was calculated by summing up individual samples on 1 cm resolution until the maximum sampling depth was reached. The pedons were assigned to a specific land cover class and the SOC content averaged for different depth intervals (30 cm, 50 cm, 100 cm, 100-200 cm, 200-300 cm, and 0-300 cm)."*

L264: what do you mean with "majority statistics"?

Majority statistics referred to the fact that the land cover type that occurred most frequently during the selected years was chosen.

To make it clear, the wording "majority statistics" was removed and sentence changed to "by identifying prevailing land cover classes within this period".

L266-268: I don't understand what you mean with this sentence

Thank you for the comment and fully agree, that this sentence is rather confusing and not of relevance here. Sentence removed.

*Results*

I find the pedon grouping confusing. You binned the data into intervals of 0-30 cm, 0-50 cm, 0-100 cm, 100-200 cm, 200-300 cm and 0-300 cm. This is not consistent as the 0-100 cm interval contains the 0-30 and 0-50 cm, but then you separate 100-200 and 200-300 cm. Why? It would be clearer to have intervals from 0-30, 30-50, 50-100, 100-200 and 200-300 cm and then have the "summary intervals" 0-100 and 0-300 cm. This way, the amount of pedons in Table 4 would also add up and it would be clearer how many pedons cover what intervals.

The intervals in both tables are now regrouped following your suggestion.

L299 and 323: use a different word instead of "bulk"

Exchanged to most in both cases.

L302: the graph also shows the distribution of the depth 0-100 cm; please describe the results of the spatial distribution of the C storage (and the same for N in the next subchapter).

*A few more additional statements added in the result and the discussion section.*

L321-322: can you back this statement statistically? I can't really confirm this in Fig 5.

*This statement is removed from L321-322, but added additional information in the section 3.3 "The data shows clear differences occurring in the more variable top meter in comparison to the rather stable second and third meter. With an exception in Non-Permafrost wetlands where the TN is more variable below 100 cm depth, which results from only 2 stratigraphy different available pedons where TN data is available (Table 7)."*

L342: rename this subchapter and describe the data visualized in Fig. 5

*Chapter title renamed to "Soil stratigraphies" and additional description to several other classes added.*

L347-348: what is the relevance of this information?

*We removed this sentence from the manuscript.*

*Discussion*

I would recommend to restructure the discussion to better follow the two study aims. It feels like the first paragraph can be left out as it repeats parts from introduction and methods.

*Discussion partly restructured, rewritten and first paragraph greatly shortened.*

L376: rephrase and clarify "within each other's error estimates

*Sentence rephrased to: "Although our values are a bit lower than their estimates, they are within each other errors."*

L377: "in comparison" does not fit to the sentence; I suggests to move this sentence as more of an outlook

*Agree and this part is now in the last section of the discussion.*

L384-388: this paragraph is very general and is quite similar to the text in the methods. Instead, really discuss the actual data.

*Following the comment from reviewer 2, this paragraph is now removed. The focus to test the database is on SOC and TN data. Several other parameters are only mentioned as part of the data, but not discussed.*

L387: reformat d13C; with "locates the areas… vulnerable to permafrost degradation" don't you mean the organic matter vulnerable to decomposition? Or can you please explain how you can define vulnerable areas with the δ13C and C/N values?

*Section removed, see above.*

L397-400: this feels a bit awkward, as you chose the surface areas from the land cover map for a reason and now you say your area is wrong?

*Thank you for the understandable comment. We choose the ESA product because it offers a high resolution land cover product on a circumpolar scale. Even it is a great*

product, it has it's limitations and unfortunately a relatively low accuracy in the very important for SOC and TN natural and semi-natural aquatic vegetation class. We conclude, that this is partly the reason for our lower estimate. But we also pointing out several other underrepresented areas which additionally increase the uncertainties. At the same time, this is a point based database which can be simply extended with additional sampling sites or to different land cover products.

*Conclusion*

L409-411: first start to mention the actual SOC estimates and then you can mention that this is lower than previous studies (although not significantly?). The part about the wetlands is not needed here I think.

Restructured and the wetland part removed.

*References*

The notation of the DOI in the references is not consistent: mostly it is written as https://doi.org/10... (which is the correct way), but sometimes it is written as doi.org/10..., doi:10... or DOI:10...

DOI's are now updated using the correct way.

Hugelius 2012: move year to the end

Moved to the end.

Kracht and Gleixner (2020): DOI is missing

DOI added

*Figures and tables*

Figure 1: source of map should be Natural Earth Data

Source text "Made with Natural Earth" is the suggested way from their homepage.

Figure 1, 3 and 4: add a space between the degree sign and the direction

Unfortunately, this is a presetting in Esri's ArcMap program and I can not change that.

Figure 3 and 4: add labels to figure panels (a) and (b)

Labels "a" and "b" added to both figures

Figure 5: write parameters in the caption

All the parameters added to the caption as also added labels for the figure panels.

Table 1: add degree sign and direction for the longitude and latitude

Degree sign and direction added as suggested

Table 4 and 6: add unit of depth

Units of depth added in both tables.

*Supplements*

Please add more information to the caption of Figure S1 to explain what information is recorded for every sample.

Additional information in the caption added as suggested.

*Formatting*

Please make sure to check the journal's guidelines on figure content and mathematical notation and equations (e.g., spaces between number and unit, units written exponentially), powered by

Thank you for the comment. Went through the document and corrected the typos such as spaces, etc.

We thank reviewer #1 for the constructive comments, which helped to improve our manuscript and hope we addressed all the questions raised by the reviewer.

**Response to Kristen Manies**

**Major comments**

This manuscript describes the data collection methods for over 6500 soil samples and uses these data to provide C and N stock estimates for the circumpolar permafrost region. These data are very useful and important. The Introduction and Results are presented well. However, I do think that the Discussion section is missing a paragraph discussing the caveats of the data. This information is only briefly mentioned in the Conclusions (line 415) and needs to be discussed more in depth in the Discussion portion of the paper. As stated in line 415, their data are concentrated in non-North American locations, such that a more complete picture could be obtained by combining their data with other datasets. In addition, the dataset only contains one high alpine site, so this ecosystem type is underrepresented. I don't expect this manuscript to do analysis beyond what is presented here, but I do think it's important to be clear about the limitations of their data and what next steps (i.e., combining with other datasets) could be taken to expand our understanding on this important information.

Dear Kristen Manies. Thank you very much for all your valuable comments to help to improve our manuscript. We fully agree, that the database has several really underrepresented areas and classes which are problematic. And it would benefit from combining this data with other available sources. As suggested, a section was added in the discussion pointing out again the issue with underrepresented sites and the benefits of combing this dataset.

In addition, there were many times when I was reading the methods that I had unanswered questions regarding specific methodology and/or how their methods might impact data quality. I would like to see many, if not all, of the following questions answered, such that others who want to use the data truly understand how it was collected. Areas where I had questions include (line #: question):

119: Do you mean that the following field descriptions were classified as wetlands? Also, why is "mineral" a wetland?

> We used the Canadian system to classify the different wetland classes in the field, including the one for mineral wetlands. All the following classes were then grouped in the Tier I class wetland (organic, mineral, seasonal, permanent, ombrotrophic and minerotrophic wetlands) To make it more clear, we made some changes in the text including a new reference.

124: I don't understand what your reason is (as no reason was stated in the previous sentence).

> Thank you for the comment. I changed the sentence to "The Tibetan permafrost region was also excluded from our estimates as none of the sampling sites originated from that area".

127: How did you define the Yedoma region? I don't think that this classification is something you can determine with site photos.

> The Yedoma extent was defined by Strauss et al. (2017), where the area was overlaid on the ESA's land cover product and constrained to the Northern Hemisphere permafrost region by Obu (2021). To clarify, text changed to "*The land cover class Yedoma is defined as areas in Siberia, Alaska, and Yukon underlain by late Pleistocene ice-rich syngenetic permafrost deposits. We used the spatial extent for the Yedoma domain from Strauss et al. (2017) which occupied an area of 570,000 km$^2$ from here ESA CCI land cover product*" as additionally described in the section 2.4.

135: How many soil descriptions per site usually?

> Added following text to the sentence: "with on average 37 sampling sites per study area"

145: Does the top organic layer mean all organic soil? Or does it mean organic soil to a certain depth? Or organic soil to a certain (estimated) bulk density?

> Rephrased to "Organic layer (OL)" as we mean the all organic soil.

146: If a steel pipe was not used, how was permafrost soil sampled?

> Permafrost was always sampled using a steel pipe except in Greenland where an Earth Auger was used.

[It would probably be helpful for the reader if the paragraphs from line 145 and 152 were combined. There would be less duplication and some of my questions raised reading the 1st paragraph were answered when reading the 2nd paragraph.]

> Thank you for this comment and fully agree that a combined paragraph makes more sense. Paragraph combined and rewritten, partly shortened but also added additional information which you asked for below and above.

156: Was the active layer never deeper than 50 cm? If so, how was the deeper active layer sampled? There must have been areas where the organics were deeper than 50 cm, especially in the wetlands. How were these soils sampled?

> Additional explanatory text added "Deeper unfrozen soil layers were sampled using a steel pipe" as of course, most sites had a much deeper active layer thickness than 50 cm.

156: The way this sentence is currently written it sounds like only in the "few cases from natural exposures" were the horizonal sampling rings used. But, according to Figure 2, this is the method used for the entire active layer. This sentence needs to be rewritten to clarify this point as well as include the information requested below.

> Thank you for again for this observant point. The horizontal sampling referred to permafrost sampling in exposures where the steel pipe was hammered in horizontally. Sentence moved to the right place of permafrost sampling.

157: I'm also worried about the sampling that happened at fixed depth intervals. How frequent were these intervals (every 5, 10, 20 cm)? Could you have missed changes in soil horizons (and thus bulk density and C concentration, affecting your C stock values) by sampling this way?

> Thank you for that comment. Usually, the frequent sampling interval of the active layer was each 10 cm with a 60 mm in diameter sample ring. So only 40 mm of pedon wall was left to stabilize the pedon. In some cases, where the soil texture was very sandy and the sampled pedon wall not stable, sapling interval had to be increased. However, minor changes in soil horizons could have been missed but the sampling intervals were dense enough and should not affect the C concentration overall. We added additional text to the manuscript.

158: I don't understand what "emphasis" means here. Or what was done when there was a lot of spatial variability within a soil pit. I think additional descriptions (i.e., depths were measured every 10 cm on the photograph and then averaged) or an example is needed here.

> As also commented by reviewer 1, this sentence is not adding any relevant information as all the sampled samples were treated similar for calculating the depth, and therefore this part is now removed.

For clarity to the reader, maybe describe the normal way you measured the active layer. Then give the details about the special cases (natural exposures, spatial distribution).

> Following information added "*Active layer thickness was measured at each location using a graduated steel probe or measuring tape in excavated soil pits.*"

170: Please clarify that the length of this pipe was measured each time it was used, so that the bulk density measurements are accurate.

> As pointed out by reviewer 1, this part is not of relevance and is now removed from the manuscript. But to clarify, every time the pipe had to be cut, it was of course remeasured and new marked.

185: How were you able to do hand manipulations on the frozen sections? Did you let part of the sample thaw and then test for soil texture? Also, if you are taking these subsamples out for texture analysis, did you return that subsample to the bag so that the weights remained accurate?

> Really good point. Yes, subsamples were thawed which happens usually within a minute and hand analyzed. And of course, returned back to the sample bag. This information is now also added in the text.

213: It'd be nice to have a few more details about how the determination of the presence of inorganic C was done. For example, were they chosen by eye? Or if the sample fell a certain percentage off the 1:1 line between LOI550 – LOI950?

> Additional information added. Inorganic carbon was only high at 2 sites, where the samples were treated with HCl prior the EA analysis. Following information in the text: "*If LOI950 following Heiri et al. (2001) indicated presence of inorganic carbon with > 1%, samples were acid treated (Ny Ålesund, Norway; Aktru, Altai mountains, Russia) with hydrochloric acid prior to determination of TOC.*".

217: Since you are using LOI data to predict C for some samples, I'd like to know a) the percentage of samples for which this predication was used (i.e., no C data available) and b) how good the fit of this relationship for these data are.

> Added additional information about the % of samples where %C was only known from LOI and added the fit between LOI and EA which is $R^2$ of 95%. Following text now in the manuscript: "*To estimate the organic carbon % (OC %) for samples where only LOI was available (44 % of samples), a polynomial regression model ($R^2$=95%) was performed between LOI550 and OC % from EA on samples for which both analyses were available at study area level*".

237: I am confused why you took three organic samples when only one was used for C stock measurements. I'm assuming that you only used OL1 because it matched with the rest of the soil profile. Maybe clarify that the other two samples were taken to quantify the variability of this layer (if this is the correct reason) and are available as data for others but aren't considered in these results.

> Similar question was raised by the other reviewer. In most cases we did collect additional organic layer as the differences can be significant. However, to avoid misunderstanding and as you correctly pointed out, we used only OL1 for the calculation as it matches the soil profile. Therefore, other OL samples were now removed from the text as they were not used anyway.

**Additional, but minor suggestions, are as follows (preceded by the line number):**

37: Are you missing a verb here? "to be 380 Pg"?

> Thank you, indeed the verb was missing. Added as suggested.

38: I found this sentence a very confusing.

> I removed this sentence as it is not relevant for the abstract.

41: What is the difference between the 2 datasets? It would be nice to have this detail, so readers know which link to use.

> Added information about the content in the "Data access" section.

66: misspelling "volumetric water content for organic soil"

> Changed

68: The word "cover" before "stones and boulders" initially made me think you were looking at those data as a percent cover. Consider removing this word (maybe need to say percentage of stones and boulders?).

> Agree about the good comment and exchanged the "cover" with "percentage of" as suggested

105: I found these two sentences confusing and think would be more understandable if they were a) a part of section 2.1 and rewritten a bit. For example, "All sites were classified with Tier 1 descriptions using field descriptions and, where possible, assigned a more detailed (Tier 2) description."

> Agree with your comment. I moved these two sentences to the next section and rewrote following your suggestion.

120: It might be clearer if you say something like "Where the permafrost status within the top 2 m of a site was known, a Tier 2 status was assigned."

> Rewrote this part

165: Please revise – you say earlier in this sentence that these soils weren't always frozen.

> Revised

191: I was confused why this information was presented, especially since I didn't see this information discussed in the results or presented in the datasets.

> Same issue was pointed out by reviewer 1. Fully agree and this part is removed from the manuscript as not further discussed.

Figure 2 is very nice.

> Thank you, I will forward this to the responsible person!

214: If you place the information that these regressions were done for each study area in this sentence readers won't be left wondering (as I was) until they read on.

> Done

218: Aren't C:N ratios usually based on percentages of these elements, not weights?

> Thank you for the comment, you are absolutely right. I removed the "weight" from this sentence.

220: Simpler to say "more decomposition"?

> Revised to your suggestion.

222: This ratio? I'm confused what "this" is referring to.

> Text revised to make it more understandable.

270: I'm confused by the words "indicated by permafrost area". I don't understand what this phrase is clarifying for the previous statement "but not the actual area underlain by permafrost"?

> Thank you for the comment. The words "indicated by permafrost area" are now removed, as they were referring to a previous version.

271: Simplify to "This dataset"?

> Revised as suggested

273: I suggest you move this paragraph to right after the paragraph on line 260 that introduces the ESA database.

> Moved this paragraph up as suggested

283: I think the first Tier mentioned should really be "Tier I".

> True, done as suggested.

286: Better to put the equation here?

> Moved the equation up.

346: I don't see Yedoma tundra (yellow line) on the Figure 5 panels for the silt+ clay nor sand panels. I also see a lot of variability in the Non-permafrost wetlands for many data types, so you may want to also mention this class.

> Figures adjusted as also additional text explaining the non-permafrost behavior.

355: The scientific communities don't have high spatial resolution, the dataset does.

> Sentence structure changed and moved the "scientific communities" to the end.

358: I think this sentence is a better topic sentence, with the sentences following this sentence explaining how it's better than what previously existed.

> Fully agree that the location of this sentence was not optimal. Moved up to be the introductory sentence for the discussion section.

374: Despite? Or because of different upscaling approaches? I find this and the following sentences to be confusing/too wordy. I think you need to focus on the points: although your values are a bit lower than their estimates, they're within each other errors. You used different upscaling approaches, which could be the cause of some of these differences. Your upscaling approach was chosen because…

> Thank you for the comment and the suggestions. This section is now rewritten following your suggestion.

382: I found this wording confusing. Maybe "estimate of 66 Pg (+/- 35 Pg) by Harden et al."?

> Changed to as suggested.

384: If you're going to have this paragraph on C:N ratios in here I think you need to discuss your results more (i.e. how they vary with land type, etc.). Right now it's just saying what you already said at line 218. There are other data you don't discuss. Maybe the focus of this paragraph should be about the other data available in this dataset and what their uses could be? Otherwise, I'd delete this paragraph.

> Thank you for that comment. I decided to remove this paragraph as the main focus is on SOC and TN storage and the used land cover product. Also as you mentioned, several other variables are not discussed as well which is not the scope of this manuscript.

390: I think "although" fits better at the beginning of the sentence as it's currently written.

> Changed to "although".

393 & 395: This what? Make sure to follow the word "this" with a noun so readers don't get confused about the subject you are discussing.

> Parts rewritten according to suggestion.

395: I don't think you need to say "in this study" here.

> Deleted

396: I'm not sure "throughout" is the appropriate word here since you're only discussing wetland classes. I suggest deleting it.

> Deleted

397: It might be clearer to say "exchange the ESA wetland areal coverage for the values in Hugelius". Also, you give us your updated estimates, but please remind us how those relate to the other estimates and what those values are. (Otherwise I have to reread the paper to find them.)

> Changes made as suggested.

401: I think your argument needs to be that you present a more complete dataset in regard to variables used to parametrize models. Because other data sets have similar data, maybe just not to the completeness you do.

> Changed as suggested.

We thank reviewer Kristen Manies for the constructive comments, which helped to improve our manuscript and hope we addressed all the questions raised by the reviewer.

---

## Author Response (AR2)

**Dear Reviewer 1 and Reviewer 2.**

**Thank you again very much for your comments and suggestions to improve our manuscript. Here is the respond to all your comments point by point:**

**Response to the anonymous Referee #1:**

**Major comments**

I have looked over your replies and the revised manuscript. I have a few questions about comments that I feel were not fully incorporated, and some more minor comments.

Especially, in my previous review, I advised you to describe and discuss the "other soil parameters" (C/N ratio, δ13C, BD, volumetric fractions and grain size fractions). You stated that the description and discussion of these parameters are beyond the focus of this paper and therefore only the data are included. But it does not add anything to report data that are not described and discussed and if not done so, you will have to take them out completely. You could publish them in a data repository, but they don't belong in the manuscript in this way. You should then also take out the description of the parameters in the methods section. However, I think it would be valuable to describe at least the C/N ratios and possibly the δ13C.

> Dear Reviewer 1, thank you for the comment. I do agree the C/N ratios and the isotopes are an important contribution of this paper. Therefore, we added an additional section in "Results" which was further discussed in the "Discussion" section. Also, added more information in the "Result" "Soil stratigraphies" part about the dependence of DBD with the mineral fraction and the Yedoma classes.

You do use the coarse fraction for the correction in Equation 1, so you could have those data in the supplements.

> Agree that this is important data, but this is exactly the dataset which accompanies the paper in the data repository. This data (n=6500) is available on sample level, same as for C%, TN%, and so on.

In the discussion, you compared the total SOC and TN store to previous estimates and explained why the numbers might be different. However, it would be very valuable to compare the SOC and TN densities to previous studies, because they are independent from the mapping. When these are different from other studies, the density values might be part of the answer to why the SOC and TN store are different in your estimates and not just the mapping.

> Thank you for that interesting point. Agree that SOC and TN densities are very important. This is also, why the accompanied dataset has this information for each available sample. Looking at our data, there is a good correlation between SOC and TN density of $R^2$=0.8. However, looking into other recent studies, which we used with circumpolar SOC estimates, no such data is available. For example, Mishra's et al. estimate is modeled based on 2700 soil profiles. But even in the supplementary material only the mean and total SOC concentrations are given with very different classes. While TN data is hardly available at all.

**Minor comments:**

L24: please add "land surface area" or something similar to "Northern Hemisphere" (as you say in L55)

> Thank you, indeed good to clarify this in the beginning. Changed as suggested.

L30-31: what do you mean with "turnover times"? Please remove "soil texture" here when not describing and discussing these parameters

> Removed as not discussed. while

L45: remove "in turn" (because you use it in the next sentence as well)

> Removed as suggested

L61-64: remove parameters that you don't describe and discuss

> We added additional information to several parameters in the text. Also, the data is used in the used soil stratigraphies in Figure 5.

L64-65: not relevant here

> Removed as suggested.

L69: you followed my suggestion to change the word objective to aim, but in the next sentence you are still using objectives. Be consistent

> Thank you and fully agree. Now "aim" is used throughout the text.

L72: replace soil organic carbon by SOC

> Changed to SOC

L107: add space between 50 and cm

> Space in between added.

L118: do you really mean "where soil is absent", cause what did you then sample?

> I added "almost" prior completely absent. But is true that in some cases, soil sampling was not possible as with predefined equidistant intervals we ended up on e.g. exposed bedrock. If possible, we could sometimes collect a few mm thin OL layer.

L127-137: Thanks for specifying what you meant with "most sites" and "for some locations". I would suggest to rephrase it a bit such as: "A stratified sampling scheme consisting of linear … elements was used to retrieve 582 soil pedons." or rather add (n=582) at the end of the sentence. Do I understand it correctly that in the other 69 cases, the last 2 sentence of the paragraph (When sufficient time… each sampling point) apply? Please add this too: "… additional sampling (69 soil pedons) using a random…"

> Thank you for additional suggestions. Made changes as suggested.

L140: use abbreviation OL or leave out introduction of abbreviation in L138

> Decided to remove the abbreviation in L138, as not used anymore anyway.

L145, 156, 163: you mention the core was described but nothing else so it does not add any info to mention it. I suggest to remove this and only mention it where you explain it (L164)

> Thank you for that comment. Indeed repetitive information which is now removed as suggested and only kept the one in L164.

L149 and 152: add (n=X) to "at several locations"

> N added to both "several locations"

L173-174: move sentence up to before you mention the thawing and analyzing

> Sentence moved one up.

L265: in other cases, you capitalized the word Tier

> All "Tier" words are now capitalized

L271: Thanks for clarifying about the replicates. However, I still don't understand what you imply with the sentence "Replicates were only considered for pedons reaching the full depth,"

> Changed slightly the sentence to. ". Replicates were only considered for pedons reaching the full sampling depth, resulting in fewer replicates available with increasing sampling depth". What we mean is that pedons to a depth of e.g. 180cm are accounted in the CI calculation only to 0-100cm and not to 0-200, since we don't know the remaining 20cm. This is why the amount of pedons for CI calculation is decreasing with depth.

L285: the same pattern as what? And L287: do you mean in the highlighted areas?

> Sentence changed to: "*Spatially, the SOC distribution in Figure 3 is following the same pattern and highlights the largest SOC content predominantly in permafrost peatlands in Western Siberia, Russia and the Nunavut territory in Canada.*"

L331, 333, 335: don't capitalize "non-permafrost" and "permafrost", "barren"

> Changed as suggested.

L362: Maybe a better way to say that your estimates are in each other's errors is to say they are not significantly different?

> I see your point but since I can not test Harden's estimate statistically due to different statistical approaches, I prefer to keep the comparison as it is but have exchanged the word "error" to "ranges".

**Response to Kristen Manies**

Thank you to the authors for their work revising the manuscript. The changes made have made it easier to read and understand. I have some more minor comments that I think would continue to help them communicate their work (see below). I have no major comments or revision suggestions.

> Dear Kristen Manies. Thank you once again for your time and your valuable comments to help to improve our manuscript. We gladly incorporate all your additional comments and suggestions.

MINOR COMMENTS:

Lines 69-71: Consider rewording these objectives. Since the word core can also be a soil core, consider using "primary" or some other synonym. I also feel like the "Secondly," sentence is missing some words. Consider something like "In addition, we used this soils dataset and ..."

> I see your point and agree, that core is not optimal wording. Changed to "primary" as suggested. Also made changes to the "Secondly".

Line 73: I think this sentence, which is about the dataset, needs to be put right after the sentence about the dataset to make sense.
> Thank you for that comment, moved this sentence up as suggested.

Line 110: I think the detail now included here on the numerous types of wetlands is confusing. At first I read this sentence thinking these were going to be your Tier II categories. I suggest that instead you say something simpler like "Areas that met the National Wetlands Working Group (1997) definition of a wetland were classified as such."

> Agree that it might sounded confusing. Changed to your suggested version.

Line 121: What do you mean by "from here ESA CCI land cover product"?

> Thank you for finding that, it's a typo. It should have been "the". I changed it now to "*which occupied an area of 570,000 km$^2$ from the used ESA CCI land cover product.*"

Line 139: Consider "or, when this was not possible, to a depth of at least 50 cm."

> Changed to as suggested.

Line 158: This sentence doesn't make sense to me. Are you trying to say "An accurate determination of soil bulk density (BD) is crucial when calculating SOC stocks"?

> Thank you for that comment. Agree, that it doesn't make sense. Sentence changed to: "*Since the accurate determination of soil bulk density (BD) is crucial when calculating SOC, special attention was paid to accurate soil volume estimation during field sampling.*"

Figure 2: This figure still shows three organic surface blocks – this figure needs to be modified now that you only discuss sampling 1 block.

> Thank you again for that observant comment. Figure now modified with only one sampling block.

Line 186+: I find these sentences confusing as written. Consider just saying "To ensure that there was no remaining water in the organic rich and/or fine grained samples (n=3684), subsamples of ~10 g were dried again at 105 C to verify the oven dried weights. There were no noteworthy differences between samples dried at the two temperatures." I also am not sure you need the rationale for this process (lines 188 – 191). If you keep it, maybe include the reasoning in the first sentence, when you first mention the lower OD weights.

> Agree and changed the sentence to the suggested version. Also, removed unnecessary parts as not needed.

Lines 197 & 198: Instead of "burned" consider "heated to"

> Changed to "heated to"

Line 201: Do not need the word "with"

> Agree and removed

Line 206: This paragraph does not seem to fit within the methods but should instead be in the Results or Discussion sections. Or deleted.

> This paragraph is now moved to the Result section.

Line 257: Do you actually use and/or present wet bulk density? I've only ever seen bulk density calculate using oven dried soil.

> Not sure if the line number "257" is correct. However, we do use the wet bulk density to calculate for example the air fraction in the soil stratigraphy's. Also, the wet bulk density is calculated for each sample and part of data available to download.

Tables 4 & 6: What does the word "landscape" signify here? You only discuss land cover and these values appear to be for all data. I think you should delete this word (so it reads "Mean SOC storage").

> Agree that the landscape in this content does not make sense, work deleted.

Line 331: I think these sentences should be combined.

> Merged as suggested.

Line 333: I don't think you need the word "instead" here. Also, would the descriptor "consistent" be better than "rather stable"?

> Thank you, exchanged the wording to the suggestions.

Line 334: I think a more simplified statement, such as "which is due to the high organic fraction of these soils" is better.

> Changed as suggested.

Line 334: I don't think its correct to start a sentence with the word "which". It also helps the reader when you set up the next idea, so they know what's coming (more evidence, a difference, etc.). Consider something more like "In comparison, the barren has the lowest SOC and TN, as these soils

are dominated by the coarse mineral fraction." Also, these classes were not capitalized earlier. Please be consistent throughout the manuscript.

> *Indeed and thank you for that comment. Changes made as suggested.*

Line 336: I'm confused, because I would think that "these important trends" would be related to the previous sentences, but you haven't discussed water content yet. In fact, to me the trend is the relationships you've been discussing have been related to the amount of organic / mineral fractions and TOC, not the silt/clay you mention here. I think this sentence needs to be rewritten some.

> *Thank you for pointing out the confusion. Sentence is now completely rewritten to*
> *"While the stratigraphy for the Yedoma classes proves the Yedoma typical ice-rich silt sediments visible in the high silt + clay and high water/ice fraction."*

Line 372: Clearer if you say "…aquatic vegetation, which corresponds to our wetland class, is unfortunately…"

> *Changed to the suggestion*

Line 376: Sentence starting with the word "Which" appears to be incomplete.

> *Sentence rewritten and additional explanation added.*

Once again, we would like to thank both reviewers for the constructive and valuable comments and suggestions, which helped us to improve the manuscript and hope we were able to addressed all the comments.